# Compressive Sensing of Medical Images Based on HSV Color Space

**DOI:** 10.3390/s23052616

**Published:** 2023-02-27

**Authors:** Gandeva Bayu Satrya, I Nyoman Apraz Ramatryana, Soo Young Shin

**Affiliations:** 1School of Applied Science, Telkom University, Bandung 40257, Indonesia; 2Department of IT Convergence Engineering, Kumoh National Institute of Technology, Gumi 39177, Republic of Korea

**Keywords:** compressive sensing, HSV, color medical imaging, sparsity averaging, reweighted analysis

## Abstract

Recently, compressive sensing (CS) schemes have been studied as a new compression modality that exploits the sensing matrix in the measurement scheme and the reconstruction scheme to recover the compressed signal. In addition, CS is exploited in medical imaging (MI) to support efficient sampling, compression, transmission, and storage of a large amount of MI. Although CS of MI has been extensively investigated, the effect of color space in CS of MI has not yet been studied in the literature. To fulfill these requirements, this article proposes a novel CS of MI based on hue-saturation value (HSV), using spread spectrum Fourier sampling (SSFS) and sparsity averaging with reweighted analysis (SARA). An HSV loop that performs SSFS is proposed to obtain a compressed signal. Next, HSV–SARA is proposed to reconstruct MI from the compressed signal. A set of color MIs is investigated, such as colonoscopy, magnetic resonance imaging of the brain and eye, and wireless capsule endoscopy images. Experiments were performed to show the superiority of HSV–SARA over benchmark methods in terms of signal-to-noise ratio (SNR), structural similarity (SSIM) index, and measurement rate (MR). The experiments showed that a color MI, with a resolution of 256×256 pixels, could be compressed by the proposed CS at MR of 0.1, and could be improved in terms of SNR being 15.17% and SSIM being 2.53%. The proposed HSV–SARA can be a solution for color medical image compression and sampling to improve the image acquisition of medical devices.

## 1. Introduction

Medical images have been exploited by researchers for medical applications over the past decades, i.e., brain tumor detection techniques using magnetic resonance imaging (MRI) [1,2,3,4], early diagnosis of colorectal cancer using colonoscopy image [5,6,7], early gastrointestinal tract cancer diagnosis using wireless capsule endoscopy (WCE) images [8,9,10,11], cholesterol level detection using eye images [12,13,14]. These applications require massive medical images for data training. For storage and transmission, medical images require effective compression algorithms [15]. Certain medical images have complicated characteristics with the color format. As a result, efficient compression of medical images, that take color information into account, is needed [16,17].

Recently, compressive sensing (CS) was presented as a novel sampling scheme for a sparse signal with sparse signal and reconstruction schemes [18,19,20,21]. CS has been approved as a method for breaking the conventional Nyquist rate and reducing the length of a medical inquiry [22]. Furthermore, CS was used for medical image compression and compressed medical imaging (CMI) was proposed as an approach for image compression or sparse sampling which exploits the concept of CS in medical imaging [23]. The majority of CMIs only handle grayscale images, and their utilization of color images remains a challenge. In general, medical images consider red–green–blue (RGB) color space for sensor and storage purposes. However, many approaches for medical images validated the superior hue-saturation value (HSV) over RGB color space, i.e., new HSV aggregation approaches for color edge detection [24], a polyps segmentation in HSV color space [25], an enhancement in HSV effectively reducing inconsistent illumination during image acquisition [26], and detection of small colon bleeding in WCE videos [11]. A comparison of the visual representation between RGB and HSV in MRI, WCE, colonoscopy, and eye images is shown in Figure 1. However, a CS scheme with HSV color space has not yet been studied.

The CS framework consists of measurement and reconstruction steps. Generally, the measurements must be taken randomly, and different techniques have been proposed in the literature, such as variable density Fourier sampling procedures used in MRI [27,28], Gaussian model [29,30,31], and Bernouilli random matrices [21]. In addition, a universal and efficient CS by spread spectrum (SS) Fourier sensing basis was proposed [32]. In the reconstruction step, basis pursuit (BP) and BP denoising (BPDN) were proposed by Chen et al. [33]. Furthermore, sparsity averaging (SA) and reweighted analysis (RA) were proposed to improve the BPDN [34].

Motivated by an efficient HSV-based sampling in color medical images, a novel HSV-based CS of medical images is proposed in this article. To the best of the authors’ knowledge, no HSV color space has been studied for the CS of medical images in the literature. The contributions of this article are listed as follows:Proposing a new CS framework by considering HSV color space.Proposing a CS approach by exploiting SS Fourier sampling in the measurement approach.Proposing a CS reconstruction with HSV loops by exploiting SA, BPDN, and the RA enhancement for MRI, WCE, colonoscopy, and eye images.

The organization of this article is as follows. Section 2 elaborates related methods. Section 3 explains the overview of CS. The proposed HSV-based CS is described in Section 4. The experiment setup is presented in Section 5. The experiment results are presented in Section 6. Finally, Section 7 concludes this article.

## 2. Related Methods

CS was recently introduced for signal/image reconstruction, and it has shown promising results from both theoretical and engineering standpoints [15]. Natural data, medical pictures, and hyperspectral photographs are all examples of its uses. By utilizing reduced sets of measurements, a nonuniform sample from sensors was used in the CS framework [35]. Data sent through wireless networks now has challenges concerning massive data generation, storage, and transport. The Fourier transform domain was proposed for CS reconstruction of non-uniform data provided by massive sensors [35,36,37]. Furthermore, a CS technique for hyperspectral images is described using sparse tensor coding of linear and nonlinear sparse data to overcome the aforesaid problems [38].

The CSs that exploit deep learning (DL) for images were studied in an efficient manner [39]. A CS reconstruction, based on a data-driven DL and conventional CS approach, was proposed for sparse images, referred to as ADMM–CSNet [40]. A multi-scale DL-based CS was proposed as a CS reconstruction approach at the multi-scale level [41]. Moreover, the CS reconstruction scheme was proposed by exploiting generative adversarial neural networks [42,43].

Carrillo et al. proposed a SA framework of multiple wavelet dictionaries and two steps of reconstruction based on BPDN and RA for CS of magnetic resonance imaging (MRI), namely SARA [44]. An improved SARA was proposed using a new sparsity basis derived from multiple SARA bases, namely multiple-BP with RA (M-BRA) for MRI, WCE, computed tomography (CT), and colonoscopy [23]. Rahim et al. proposed CS of CT images utilizing SARA based on total variation denoising (TVDN), instead of BPDN, to minimize reconstruction time, namely TV-SARA [45]. The CS of WCE images that considers RGB color space was introduced and RGB–SARA was proposed [46]. Next, to reduce the reconstruction time, RGB–BPSA was proposed using BPDN and SA for CS of eye image [47]. In addition, RGB–TV was proposed for CS of eye images as an initial investigation of TVDN in RGB-based color images [48]. Finally, Table 1 presents the related methods. In this article, RGB–SARA [46], RGB–BPSA [47], and RGB–TV [48] were considered as the benchmark methods.

## 3. Overview of CS for Medical Images

CS is a novel sampling technique and images can be represented sparsely or compressibly in a spatial or transform domain. CS samples the image at a rate significantly lower than the Nyquist sampling rate by relying on image sparsity. Furthermore, the diverse reconstruction methods can recover the image from less compressive samples.

In this section, an overview of CS for medical images is presented as shown in Figure 2. First, a signal s with dimension n×1 is generated from a medical image. Next, the signal s is transformed to a sparse signal x using sparsity basis Ψ with dimension n×n or defined as x=Ψs. In certain sensing matrices $\Phi \in C^{m\times n}$, the compressed signal $y\in C^{m}$ is represented by m×1 linear samples and formulated as
(1)y=Φx.

According to Equation (Equation 1), one solution to recover x from y is to acquire the sparse representation $\bar{x}$ with respect to the known measurement matrix Φ. As a result, this CS reconstruction approach can be represented by a convex problem as
(2)$$\begin{array}{cccc} \; & \min_{\bar{x}}\; & \; & ∥\bar{x}{∥}_{0}\\ \; & s.t.\; & \; & ∥y-\Phi \bar{x}{∥}_{2}, \end{array}$$
where ${∥x∥}_{a}={\left(\sum_{i=1}^{m}{\left|\alpha_{i}\right|}^{a}\right)}^{\frac{1}{a}}$ is the $\ell_{a}$ norm of a vector x. The most popular way to solve the issue in Equation (Equation 2) when $\ell_{0}$ norm is substituted by $\ell_{1}$ norm is a convex problem [18].

## 4. Proposed Methodology

A novel SARA-based CS of the eye image is proposed with HSV loops, as shown in Figure 3. Firstly, a color medical image is considered as the original image and denoted by $s\in Z^{N\times N\times 3}$, where Z is an unsigned integer number. Then, HSV loops are performed on each HSV layer.

### 4.1. HSV

The conversion of RGB to HSV is presented as follows. First, the maximum intensity of RGB is determined as $I_{\max}=\max\left(R,G,B\right)$, the minimum intensity of RGB is determined as $I_{\min}=\min\left(R,G,B\right)$, and the intensity range is calculated from $I_{\max}$ and $I_{\min}$ as $I_{\mathrm{diff}\;}=I_{\max}-I_{\min}$. Next, *H*, *S*, and *V* are calculated as
(3)$$\begin{array}{cc} \; & H=\left\{\begin{array}{cc} 0 & \;\mathrm{if}\;I_{\max}=I_{\min}\\ \left(60^{\circ }\times \frac{G-B}{I_{\mathrm{diff}\;}}+0^{\circ }\right)\;\mathrm{mod}\;360^{\circ } & \;\mathrm{if}\;I_{\max}=R\\ \left(60^{\circ }\times \frac{B-R}{I_{\mathrm{diff}\;}}+120^{\circ }\right) & \;\mathrm{if}\;I_{\max}=G\\ \left(60^{\circ }\times \frac{R-G}{I_{\mathrm{diff}\;}}+240^{\circ }\right) & \;\mathrm{if}\;I_{\max}=B \end{array}\right.,\\ \; & S=\left\{\begin{array}{cc} 0 & \;\mathrm{if}\;I_{\max}=0\\ \frac{I_{\mathrm{diff}\;}}{I_{\max}} & \;\mathrm{else}\; \end{array}\right.,\\ \; & V=I_{\max}. \end{array}$$
Furthermore, the pixel range of HSV and RGB are as summarized in Table 2.

### 4.2. HSV Loop

Each HSV loop process begins with a preparation step, using a single-layer image as the input and the prepared images as the output. Pixel normalization and enforce positivity are two steps in the preparation process. Pixel normalization is a method of converting a range of pixel intensities into a normalized range of 0 and 1. Enforce positivity, on the other hand, is a method of removing negative values following the pixel normalization procedure. Figure 4a depicts a graphic representation of the preparation procedure.

### 4.3. Measurement with Spread Spectrum Fourier Sampling

For each layer, first, preparation is performed to obtain a prepared image $P\in R^{N\times N}$ as the input of the spread spectrum Fourier sampling (SSFS). In the preparation step, pixel normalization and enforce positivity are performed.

Next, in SSFS, the sensing matrix is the spread spectrum matrix and is defined as
(4)Φ=MFS,
where M, F, and S represent the mask matrix, the discrete Fourier coefficient matrix, and the spread spectrum matrix, respectively. The inverse of MFS is defined as
(5)$$F^{T}M^{T}1_{M},$$
where $1_{M}\in R^{M}$ represents a masking matrix consisting of only values of zero and one value. Last, if HSV loops are finished, then the compressed signal y is obtained.

### 4.4. SA

In SA [46,47], the sparse basis Ψ is generated from *p* multiple wavelet bases. The wavelet filter type is Daubechies (db) with level decomposition *l*. In general, the sparsity averaging basis Ψ is calculated as
(6)$$\Psi =\frac{1}{\sqrt{p}}\left[\Psi_{1},\Psi_{2},\ldots ,\Psi_{p}\right],$$
where $\Psi_{1}$ denotes db with 1 tap wavelet filter and $\Psi_{p}$ symbolizes db with *p* taps. In this article, a simple averaging basis is proposed using a db1–bd8 basis and defined as
(7)$$\Psi =\frac{1}{\sqrt{2}}\left[\Psi_{1},\Psi_{8}\right],$$

### 4.5. CS Reconstruction

The CS reconstruction problem with BPDN is modeled as
(8)$$\begin{array}{cccc} \; & \min_{\bar{s}}\; & \; & {∥\Psi \bar{s}∥}_{1}\\ \; & s.t.\; & \; & ∥y-\Phi^{†}\Psi \bar{s}{∥}_{2}\leq \varepsilon , \end{array}$$
where $\Phi^{†}$ is ad-joint operator of Φ. SARA is CS reconstruction that lies on SA, BPDN, and RA enhancement. Algorithm 1 presents the step of reconstruction using HSV–SARA. The RA enhancement to BPDN solution is a $\ell_{1}$ minimization with a reweighted technique, where a weighted $\ell_{1}$ norm replaces the $\ell_{0}$ norm. The RA is terminated according to a condition, i.e., β less than ε∈(0,1), or $i=i_{\max}$ is obtained. The RA is modeled as
(9)$$\begin{array}{cccc} \; & \min_{\bar{x}}\; & \; & {∥W\Omega_{m}^{†}\bar{x}∥}_{1}\\ \; & s.t.\; & \; & ∥y-\Phi \bar{x}{∥}_{2}\leq \varepsilon , \end{array}$$
where $W\in R^{m\times m}$ is weight matrix. Figure 4c depicts a visual representation example of HSV-SARA result.
**Algorithm 1:** HSV-SARA**Input**: Measured signal $y\in C^{m\times 1}$, Sensing matrix $\Phi \in C^{m\times n}$, and $\ell_{2}$ norm upper bound ε**Output**: Reconstructed signal $\hat{x}\in C^{n\times 1}$Generate SA basis ΨInitialization l=1;**while**l≤3**do**
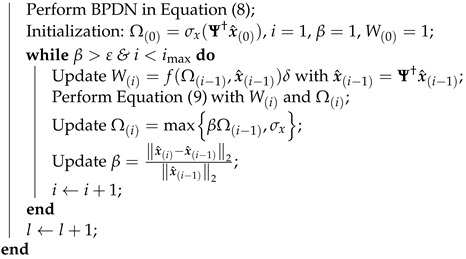


## 5. Experiment Setup

### 5.1. MRI Images

Brain tumor image data used in this article were obtained from the MICCAI 2012 Challenge on Multimodal Brain Tumor Segmentation [49,50], organized by B. Menze, A. Jakab, S. Bauer, M. Reyes, M. Prastawa, and K. van Leemput. The challenge database contains fully anonymized images from the following institutions: ETH Zurich, University of Bern, University of Debrecen, and University of Utah. The data is distributed under a Creative Commons 3.0 license: Attribution-NonCommercial (CC BY-NC).

### 5.2. WCE Images

The WCE images were obtained from the gastrointestinal tract for the analysis [8]. The detail of the WCE images are as follows: 128 images, *.JPG format file, resolution of 480×482 pixels, and RGB color space format.

### 5.3. Colonoscopy Images

The colonoscopy images were obtained from the CVC-ClinicDB database and had polyps [51]. CVC-ClinicDB is a database of frames extracted from colonoscopy videos. CVC-ClinicDB is the official database to be used in the training stages of MICCAI 2015 Sub-Challenge on Automatic Polyp Detection Challenge in Colonoscopy Videos. The use of this database is completely restricted for research and educational purposes. The use of this database is forbidden for commercial purposes. The detail of the colonoscopy images are as follows: 612 representative images, *.TIF format file, resolution of 384 × 288 pixels, and RGB color space format.

### 5.4. Private Eye Images

The private eye images from references [12,14] were used. The images were sampled from patients at TelkomMedika hospital, Bandung, Indonesia. The detail of the color retinal images are as follows: 90 images, 660×603 pixels, *.bmp format, and RGB color space.

### 5.5. CS Quality Metrics

In this section, the CS quality metrics are presented. First, the measurement ratio (MR) is a ratio between the sample number of measured signal y and sparse signal x. MR is calculated as
(10)$$\mathrm{MR}=\frac{m}{n},$$
where *n* is the sample number of sparse signal x, *m* is sample number of measured signal y, and 0<MR≤1. In this article, MR=0.1,0.2,0.3,0.4,0.5 were investigated.

Second, the signal-to-noise ratio (SNR) is a logarithmic decibel scale between the desired medical image signal s and the error of the reconstructed signal $\hat{s}$ to s. SNR is calculated as
(11)$$\mathrm{SNR}=\frac{1}{3}\sum_{l=1}^{3}20\log_{10}\left(\frac{{∥s_{l}∥}_{2}}{{∥s_{l}-\hat{s_{l}}∥}_{2}}\right),$$
where *l* denotes the image layer and ${∥\;∥}_{2}$ represents the $\ell_{2}$ norm.

Last, the structural similarity (SSIM) index is an image perceptual metric with respect to the quality loss of compression that relies on the luminance, contrast, and structure of the image. The luminance, contrast, and structure are determined as
(12)$$\begin{array}{cc} \mathrm{lum}(s,\hat{s}) & =\frac{2\mu_{s}\mu_{\hat{s}}+C_{1}}{\mu_{s}^{2}+\mu_{\hat{s}}^{2}+C_{1}},\\ \mathrm{con}(s,\hat{s}) & =\frac{2\sigma_{s}\sigma_{\hat{s}}+C_{2}}{\sigma_{s}^{2}+\sigma_{\hat{s}}^{2}+C_{2}},\\ \mathrm{struc}(s,\hat{s}) & =\frac{\sigma_{s\hat{s}}+C_{3}}{\sigma_{s}\sigma_{\hat{s}}+C_{3}}, \end{array}$$
respectively. $\mu_{s}$ is the local mean of the pixel in image s, $\sigma_{s}$ is the local mean of the pixel in image s, $\mu_{\hat{s}}$ is the local mean of the pixel in image $\hat{s}$, $\sigma_{\hat{s}}$ is the local mean of the pixel in image $\hat{s}$, and $\sigma_{s\hat{s}}$ is the cross-covariance of s to $\hat{s}$. As a result, the SSIM is calculated as
(13)$$\mathrm{SSIM}\left(s,\hat{s}\right)={\left[\mathrm{lum}\left(s,\hat{s}\right)\right]}^{a}\cdot {\left[\mathrm{con}\left(s,\hat{s}\right)\right]}^{b}\cdot {\left[\mathrm{struc}\left(s,\hat{s}\right)\right]}^{c}.$$
Let $C_{3}$ and $C_{3}=\frac{C_{2}}{2}$ and a=b=c=1 are assumed, in simply, SSIM calculated as
(14)$$\mathrm{SSIM}\left(s,\hat{s}\right)=\frac{\left(2\mu_{s}\mu_{\hat{s}}+C_{1}\right)\left(2\sigma_{s\hat{I}}+C_{2}\right)}{\left(\mu_{s}^{2}+\mu_{\hat{s}}^{2}+C_{1}\right)\left(\sigma_{s}^{2}+\sigma_{\hat{s}}^{2}+C_{2}\right)}.$$

## 6. Experiment Results

The experiment results are described to validate the analysis of HSV–SARA performances in terms of SNR and SSIM.

### 6.1. SNR Results

The SNR of HSV–SARA, RGB–SARA [46], RGB–BPSA [47], and RGB–TV [48] were compared with regard to MRs for MRI, WCE, colonoscopy, and eye images. The parameter setup was as follows: number of basis p=8 in RGB–BPSA and RGB–SARA, the level decomposition of 4 levels, and Daubechies type were fixed for this scenario.

Figure 5 shows the SNR results. HSV–SARA offered the best result and outperformed all benchmark methods. The SNR improvements were:Regarding the MRI images, RGB–SARA, RGB–BPSA, and RGB–TV were improved by HSV–SARA, with improvements of 2 dB, 6 dB, and 8 dB, respectively.Regarding the WCE images, HSV–HSV outperformed RGB–SARA, RGB–BPSA, and RGB–TV, with 2 dB, 10 dB, and 11 dB, respectively.Regarding the colonoscopy images, HSV–HSV outperformed RGB–SARA, RGB–BPSA, and RGB–TV, with 2 dB, 5 dB, and 7 dB, respectively.Regarding the eye images, HSV–HSV outperformed RGB–SARA, RGB–BPSA, and RGB–TV, with 2 dB, 6 dB, and 7 dB, respectively.

In addition, Table 3 presents the detailed SNR results with the mean and standard deviations of all medical modalities. From Table 3, HSV–SARA could improve RGB–SARA with an average SNR improvement of 11.37% for MRI, 6.69% for WCE, 7.21% for colonoscopy, and 8.14% for eye images. In general, RGB–SARA was improved by the HSV-model for all medical images with an average SNR of 8.35%. HSV–SARA could improve RGB–BPSA with an SNR improvement of 28.34%, 35.86%, 16.98%, and 5.36% for MRI, WCE, colonoscopy, and eye images, respectively. In general, the average SNR improvement of RGB–BPSA was 21.64% for all medical images. Last, the SNR of RGB–TV improved by 21.84% for MRI, 21.38% for WCE, 11.63% for colonoscopy, and 13.70% for eye images. RGB–TV was generally improved by the HSV-model for all medical images, with an average SNR of 17.14%.

### 6.2. SSIM Results

The SSIM of HSV–SARA, RGB–SARA [46], RGB–BPSA [47], and RGB–TV [48] were compared with regard to MRs for MRI, WCE, colonoscopy, and eye images. The parameter setup was as follows: number of basis p=8 in RGB-BPSA and RGB-SARA, the level decomposition of 4 levels, and Daubechies type were fixed for this scenario.

Figure 6 shows the SSIM results. HSV–SARA offered the best result and outperformed all benchmark methods. The SSIM improvements were:Regarding the MRI images, RGB–SARA, RGB–BPSA, and RGB–TV were improved by HSV–SARA, with improvements of 0.0044, 0.0398, and 0.0238, respectively.Regarding the WCE images, HSV–HSV outperformed RGB–SARA, RGB–BPSA, and RGB–TV, with 0.0234, 0.017, and 0.0428, respectively.Regarding the colonoscopy images, HSV–HSV outperformed RGB–SARA, RGB–BPSA, and RGB–TV, with 0.0114, 0.0370, and 0.0290, respectively.Regarding the eye images, HSV–HSV outperformed RGB–SARA, RGB–BPSA, and RGB–TV, with 0.0068, 0.0216, and 0.0232, respectively.

In addition, Table 4 presents the detailed SSIM results with the mean and standard deviations of all medical modalities. From Table 4, HSV–SARA could improve RGB–SARA with an average SSIM improvement of 0.4560% for MRI, 2.4521% for WCE, 1.1803% for colonoscopy, and 0.7016% for eye images. HSV–SARA could improve RGB–BPSA with an SNR improvement of 4.5053%, 1.8562%, 4.0334%, and 2.3066% for MRI, WCE, colonoscopy, and eye images, respectively. Last, the SNR of RGB–TV was improved by 2.5859% for MRI, 4.6569% for WCE, 3.1380% for colonoscopy, and 2.4880% for eye images. In general, RGB–SARA, RGB–BPSA, and RGB–TV were improved by the HSV-model for all medical images, with average SSIMs of 1.1975%, 3.1754%, and 3.2172%, respectively.

## 7. Conclusions

This article investigated the effect of HSV color space in CS of medical images. HSV–SARA was proposed, using spread spectrum Fourier sampling in the CS measurement and BPDN with reweighted analysis in the CS reconstruction. The medical image was measured by using spread spectrum Fourier sampling, while the BPDN with Wavelet-based sparsity averaging was exploited for the reconstruction. By taking advantage of the HSV color space of medical image properties, the reconstructed images responded well to the RGB-based CS. Therefore, the measurement ratio could be significantly improved. The experimental results revealed that the proposed HSV–SARA outperformed other the benchmark methods, such as RGB–SARA [46], RGB–BPSA [47], and RGB–TV [48]. For SNR results, the improvement of HSV–SARA was 8.35% for RGB–SARA, 21.64% for RGB–BPSA, and 17.14% for RGB–TV, respectively. For SSIM results, RGB–SARA, RGB–BPSA, and RGB–TV were improved by the HSV-model for all medical images, with average SSIMs of 1.1975%, 3.1754%, and 3.2172%, respectively.

## Figures and Tables

**Figure 1 sensors-23-02616-f001:**
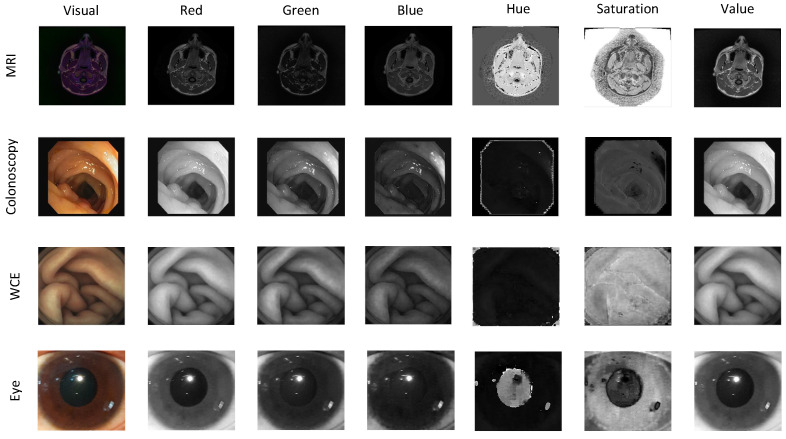
A comparison of the visual representation between RGB and HSV in medical imaging.

**Figure 2 sensors-23-02616-f002:**
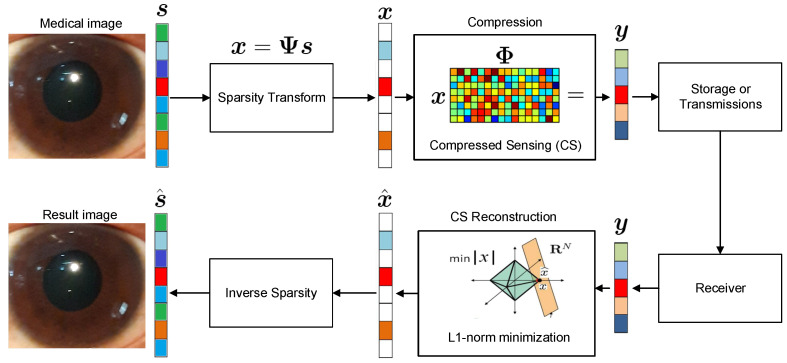
CS for medical image.

**Figure 3 sensors-23-02616-f003:**
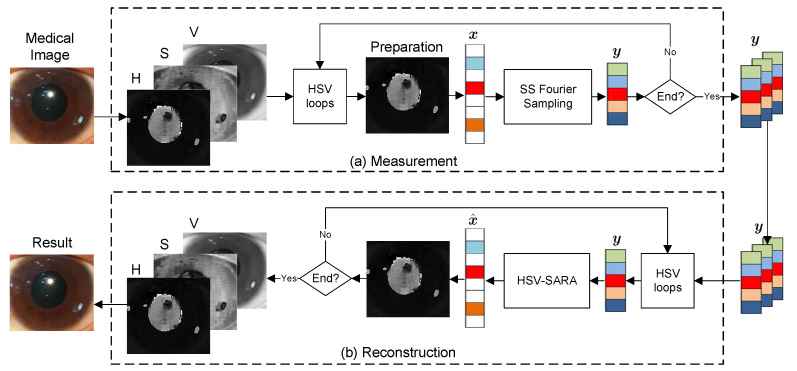
The proposed methodology.

**Figure 4 sensors-23-02616-f004:**
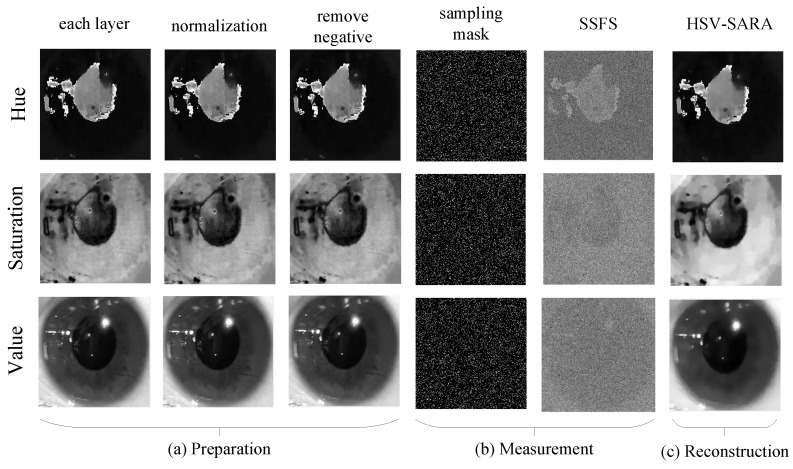
An example of visual representation in the proposed CS.

**Figure 5 sensors-23-02616-f005:**
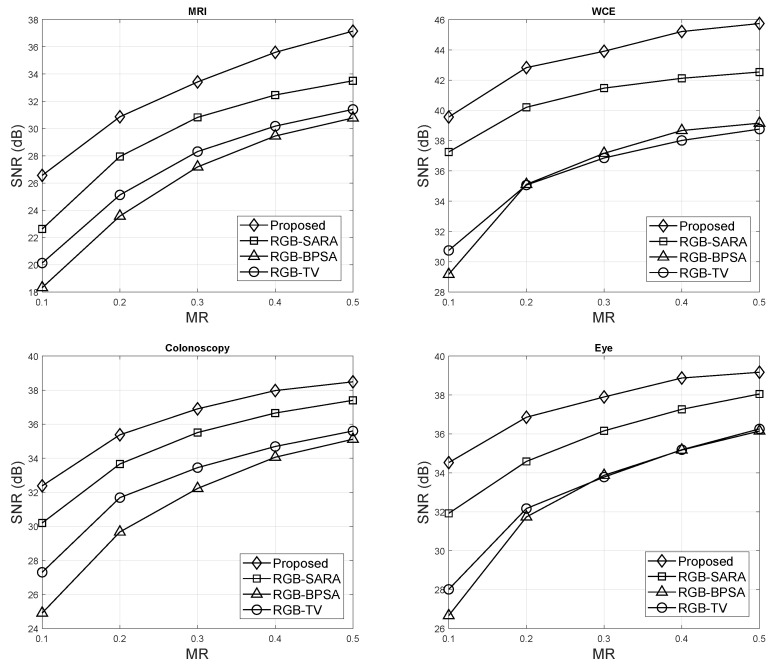
SNR results.

**Figure 6 sensors-23-02616-f006:**
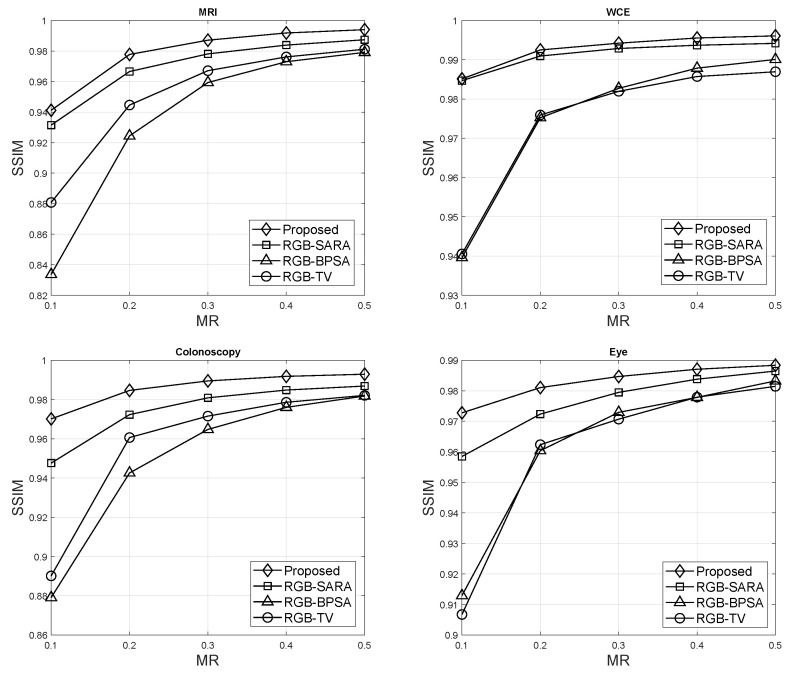
SSIM results.

**Table 1 sensors-23-02616-t001:** Related methods.

Ref	Method Name	Sparsity	Reconstruction	Medical Modality	Color
[44]	SARA	SA	BPDN with RA	MRI	Grayscale
[23]	M-BRA	Multi-SA	BPDN with RA	MRI, CT, WCE, Colonoscopy	Grayscale
[45]	TV-SARA	SA	TVDN with RA	CT	Grayscale
[46]	RGB-SARA	SA	BPDN with RA	WCE	RGB
[47]	RGB-BPSA	SA	BPDN	Eye	RGB
[48]	RGB-TV	-	TVDN	Eye	RGB
This article	HSV-SARA	SA	BPDN with RA	MRI, WCE, Colonoscopy, Eye	HSV

**Table 2 sensors-23-02616-t002:** Color space of HSV and RGB.

	H	S	V	R	G	B
Min	0	0	0	0	0	0
Max	360	100	100	255	255	255

**Table 3 sensors-23-02616-t003:** SNR results.

Medical Modality	MR	Proposed	RGB-SARA [46]	RGB-BPSA [47]	RGB-TV [48]
	0.1	26.56±2.17	22.62±2.88	18.32±2.63	20.12±2.63
	0.2	30.86±1.38	27.94±2.54	23.57±2.54	25.12±2.49
MRI	0.3	33.42±0.96	30.82±2.17	27.18±1.98	28.31±1.85
	0.4	35.59±0.65	32.46±1.88	29.45±1.71	30.17±1.56
	0.5	37.15±0.68	33.50±1.61	30.77±1.33	31.40±1.37
	0.1	39.56±1.68	37.25±0.76	29.16±2.79	30.74±4.47
	0.2	42.82±1.09	40.20±0.75	35.10±2.27	35.06±1.55
WCE	0.3	43.90±1.02	41.47±0.82	37.16±1.71	36.85±1.55
	0.4	45.21±1.18	42.12±0.83	38.67±1.69	38.01±1.28
	0.5	45.73±0.95	42.53±0.83	39.15±1.51	38.76±1.52
	0.1	32.37±1.77	30.19±2.85	24.90±3.02	27.30±6.40
	0.2	35.37±1.58	33.66±2.48	29.66±2.89	31.68±2.56
Colonoscopy	0.3	36.90±1.44	35.50±2.13	32.22±2.44	33.44±2.23
	0.4	37.97±1.55	36.65±1.80	34.05±2.01	34.68±1.80
	0.5	38.48±1.42	37.40±1.56	35.11±1.54	35.59±1.59
	0.1	34.52±1.13	31.91±2.99	26.66±2.76	28.01±5.24
	0.2	36.85±1.21	34.58±3.31	31.73±2.58	32.16±2.32
Eye	0.3	37.89±1.22	36.15±3.32	33.87±2.58	33.78±2.47
	0.4	38.87±1.25	37.26±3.19	35.17±3.09	35.18±2.44
	0.5	39.16±1.15	38.05±3.02	36.14±2.73	36.25±2.55

**Table 4 sensors-23-02616-t004:** SSIM results.

Medical Modality	MR	Proposed	RGB-SARA [46]	RGB-BPSA [47]	RGB-TV [48]
	0.1	0.936±0.024	0.931±0.047	0.833±0.059	0.880±0.063
	0.2	0.973±0.009	0.966±0.030	0.924±0.037	0.944±0.033
MRI	0.3	0.982±0.005	0.978±0.020	0.959±0.022	0.967±0.018
	0.4	0.987±0.003	0.983±0.013	0.973±0.014	0.976±0.012
	0.5	0.989±0.002	0.987±0.008	0.979±0.009	0.981±0.008
	0.1	0.985±0.007	0.984±0.003	0.939±0.050	0.940±0.113
	0.2	0.992±0.002	0.990±0.001	0.975±0.028	0.975±0.012
WCE	0.3	0.994±0.001	0.992±0.001	0.982±0.018	0.981±0.012
	0.4	0.995±0.001	0.993±0.001	0.987±0.014	0.985±0.003
	0.5	0.996±0.000	0.994±0.001	0.990±0.003	0.986±0.009
	0.1	0.970±0.012	0.947±0.022	0.879±0.058	0.890±0.193
	0.2	0.984±0.006	0.972±0.011	0.942±0.037	0.960±0.026
Colonoscopy	0.3	0.989±0.003	0.980±0.006	0.964±0.029	0.971±0.024
	0.4	0.991±0.002	0.984±0.003	0.975±0.023	0.978±0.011
	0.5	0.992±0.001	0.986±0.003	0.981±0.011	0.982±0.010
	0.1	0.972±0.007	0.958±0.020	0.912±0.041	0.906±0.155
	0.2	0.981±0.006	0.972±0.015	0.960±0.016	0.962±0.012
Eye	0.3	0.984±0.004	0.979±0.011	0.972±0.010	0.970±0.015
	0.4	0.987±0.003	0.983±0.008	0.977±0.020	0.977±0.007
	0.5	0.988±0.002	0.986±0.006	0.983±0.007	0.981±0.013

## Data Availability

The data presented in this study are available from the corresponding author upon request.

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
