# Peer review of "Compressive Sensing of Medical Images Based on HSV Color Space"

_sensors, 2023, doi:10.3390/s23052616_

Round 1

Reviewer 1 Report

A compressive Sensing of medical image based on HSV color which divided into two steps: measurement and reconstruction is presented in this paper.

There is a verbiage in the use of symbols,

In line 49, the authors stated that color feature of color medical image is more efficient in HSV color model. But no clear justification  to this claim is found

The paper language need to be revised.. for example in line 84, "to improved " must be replaced to : to improve, and so on

Although the processing reconstruction time is reduced to 20 second (should give more discussion), as authors presented, but, the nobility of the proposed method has still poor

Reviewer 2 Report

This paper proposes a novel Compressive sensing framework for medical imaging based on Hue Saturation Value (HSV). The authors proposed a spread spectrum Fourier sampling, and three CS reconstruction methods.   There are some minor issues to be addressed by the authors:   Authors should review the english language in the text.   Authors should better explain the novel spread spectrum Fourier sampling in the introduction.   In the last line of Section 3, after Eq. (2), the equation of reconstruction could be included as an equation; for example, a new Eq. (3).   In the second paragraph of Subsection 4.2, authors defined the sensing matrix as the spread spectrum matrix. However, in the second line after the Eq. (4), they stated that matrix A represents the spread spectrum matrix. Could you better explain the distiction of matrices \psi and A?

Are the results the average of some number of simulation runs?

Reviewer 3 Report

This paper proposes a CS method for medical images based on HSV color space. Some comments are shown as follows.

1) The typesetting should be improve, especially the positions of the figures and tables.

2) How to implement "CS measurement" using a sensor/camera? What is the main advantage of the CS process?

3) The improvement of the CPU time is recommented to be decribed using a relative value, such as 5%.

4) Why using three different methods for the three types images? 

5) It is suggested to add a discussion section to explain why the proposed method can improve the image quality and reduce the time? 
